# COMPOSITIONAL EMBEDDINGS: JOINT PERCEPTION AND COMPARISON OF CLASS LABEL SETS

**Zeqian Li and Jacob Whitehill**
Department of Computer Science
Worcester Polytechnic Institute
Worcester, MA 01609, USA
{zli14,jrwhitehill}@wpi.edu

## ABSTRACT

We explore the idea of *compositional set embeddings* that can be used to infer not just a single class, but the *set* of classes associated with the input data (e.g., image, video, audio signal). This can be useful, for example, in multi-object detection in images, or multi-speaker diarization (one-shot learning) in audio. In particular, we devise and implement two novel models consisting of (1) an embedding function $f$ trained jointly with a "composite" function $g$ that computes *set union* operations between the classes encoded in two embedding vectors; and (2) embedding $f$ trained jointly with a "query" function $h$ that computes whether the classes encoded in one embedding *subsume* the classes encoded in another embedding. In contrast to prior work, these models must both *perceive* the classes associated with the input examples, and also *encode* the relationships between different class label sets. In experiments conducted on simulated data, OmniGlot, and COCO datasets, the proposed composite embedding models outperform baselines based on traditional embedding approaches.

## 1 INTRODUCTION

Embeddings, especially as enabled by advances in deep learning, have found widespread use in natural language processing, object recognition, face identification and verification, speaker verification and diarization (i.e., who is speaking when (Sell et al., 2018)), and other areas. What embedding functions have in common is that they map their input into a fixed-length distributed representation (i.e., continuous space) that facilitates more efficient and accurate (Scott et al., 2018) downstream analysis than simplistic representations such as one-of-$k$. Moreover, they are amenable to one-shot and few-shot learning since the set of classes that can be represented does not depend directly on the dimensionality of the embedding space.

Previous research on embeddings has focused on cases where each example is associated with just one class (e.g., the image contains only one person's face). In contrast, we investigate the case where each example is associated with not just one, but an entire *subset* of classes from a universe $\mathcal{S}$. The goal is to embed each example so that questions of two types can be answered (see Figure 1(a)): (1) Is the set of classes in example $x_a$ equal to the *union* of the classes in examples $x_b$ and $x_c$? (2) Does the set of classes in example $x_a$ *subsume* the set of classes in example $x_b$? Importantly, we focus on settings in which the classes present in the example must be perceived automatically.

We approach this problem using *compositional set embeddings*. Like traditional embeddings, we train a function $f$ that maps each example $x \in \mathbb{R}^n$ into an embedding space $\mathbb{R}^m$ so that examples with the same classes are mapped close together and examples with different classes are mapped far apart. Unlike traditional embeddings, our function $f$ is trained to represent the *set* of classes that is associated with each example, so that questions about set union and subsumption can be answered by comparing vectors in the embedding space. We do not assume that the mechanism by which examples (e.g., images, audio signals) are rendered from multiple classes is known. Rather, the rendering process must be learned from training data. We propose two models, whereby $f$ is trained jointly with either a "composition" function $g$ (Model I) that answers questions about set union, or a "query" function $h$ (Model II) that answers question about subsumption (see Figure 1(a)).

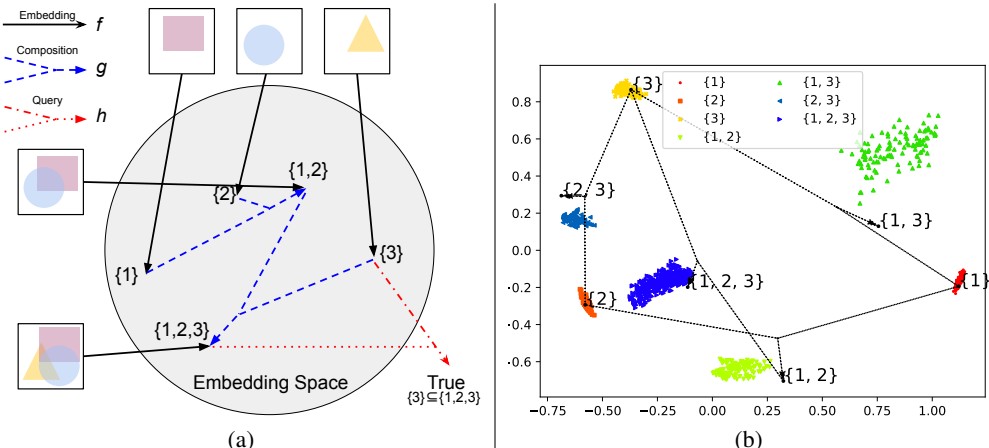

Figure 1: **(a)**: Overview of the paper: embedding function $f$ is trained jointly with either the composition function $g$ or the query function $h$. In particular, the goal is for $g$ to "compose" the embeddings of two examples, containing classes $\mathcal{T}$ and $\mathcal{U}$ respectively, to approximate the embedding of an example containing classes $\mathcal{T} \cup \mathcal{U}$. **(b)**: 2-D projection of the embedding space from Experiment 1 on test classes and examples not seen during training (one-shot learning). Function $g$ composes two embeddings (two arrow tails) and maps the result back into the embedding space (arrow head). To substantial if imperfect degree, the embedding space is compositional as described in (a).

To our knowledge, this computational problem is novel. We see at least two use-cases: (1) Speaker recognition and diarization (i.e., infer who is talking within an audio signal) with multiple simultaneous speakers: Given an audio signal containing speakers who were not part of the training set and who may be speaking simultaneously, and given one example of each person speaking in isolation (one-shot learning), infer which *set* of speakers is talking. (2) Multi-object recognition in images: Given just the embedding of an image $x_a$, answer whether $x_a$ contains the object(s) in another image $x_b$. Storing just the embeddings but not the pixels could potentially be more space-efficient. Because of the novelty of the problem, it was not obvious to what baselines we should compare. When evaluating our models, we sought to assess the unique contribution of the *compositional* embedding above and beyond what a "traditional" embedding could achieve. Hence, we created baselines by endowing a traditional embedding with some extra functionality to enable it to infer label sets.

**Modeling assumptions and notation**: For generality, we refer to the data to be embedded (images, videos, audio signals, etc.) simply as "examples". Let the universe of classes be $\mathcal{S}$. From any subset $\mathcal{T} \subseteq \mathcal{S}$, a ground-truth rendering function $r : 2^{\mathcal{S}} \to \mathbb{R}^n$ "renders" an example, i.e., $r(\mathcal{T}) = x$. Inversely, there is also a ground-truth classification function $c : \mathbb{R}^n \to 2^{\mathcal{S}}$ that identifies the label set from the rendered example, i.e., $c(x) = \mathcal{T}$. Neither $r$ nor $c$ is observed. We let $e_{\mathcal{T}}$ represent the embedding (i.e., output of $f$) associated with some example containing classes $\mathcal{T}$.

**Contribution**: To our knowledge, this is the first paper to explore how embedding functions can be trained both to *perceive* multiple objects in the example and to *represent* the set of detected objects so that set operations can be conducted among embedded vectors. We instantiate this idea in two ways: Model I for set union ($f$ & $g$) and Model II for set containment ($f$ & $h$). By evaluating on synthetic data, OmniGlot handwritten image data (Lake et al., 2015), as well as the COCO dataset (Lin et al., 2014), we provide a proof-of-concept that "compositional set embeddings" can work.

## 2 RELATED WORK

**Embeddings**: We distinguish between two types of embeddings: (1) "Perceptual" embeddings such as for vision (Facenet (Schroff et al., 2015)) and speech (x-vector (Snyder et al., 2018)) where each class (e.g., person whose voice was recorded or face was photographed) may contain widely varying examples across speech content, facial expression, lighting, background noise, etc. (2) Word embeddings (word2vec (Mikolov et al., 2013), GloVe (Pennington et al., 2014)) where each class

contains only one exemplar by definition. Within the former, the task of the embedding function is to map examples from the same class close together and examples from other classes far apart. This often requires deep, non-linear transformations to be successful. With word embeddings, the class of each example is already clear and does not need to be inferred; instead, the goal is to give the embedded vectors geometric structure to reflect co-occurrence, similarity in meaning, etc.

**Compositional embeddings**: Since at least 30 years, AI researchers, cognitive scientists, and computational neuroscientists have explored how the embeddings of multiple elements could be combined to reflect relationships between them or higher-level semantics. However, almost all this work was based on word embeddings, in which perception was not necessary. Some early work investigated how the grammatical structure and/or semantics of an input sentence can be represented (Pollack, 1989) in an efficient manner in neural networks, and how such a network could be trained (Elman, 1993). Given the advent of word embeddings, deep NLP architectures can combine the word-level semantics, as represented by the embeddings of the individual elements of an input sentence, to infer higher-level attributes, e.g., sentiment (Nakov et al., 2016). Recent work has investigated to what extent contemporary recurrent neural networks can generalize to understand novel sentences (Lake & Baroni, 2017) consisting of known words. Also, in the NLP domain, Joshi et al. (2018) developed compositional pairwise embeddings that model the co-occurrence relationships between two words given their common context. Probably the most algorithmically similar work to ours is by Lyu et al. (2019) on compositional network embeddings: the goal is to predict whether two new nodes in a graph, which were not observed during training, are adjacent, using node-based features as predictors. In their approach, two embeddings are used: one to embed the node-based features, and another to aggregate these embedded features into a secondary embedding space. Structurally, their work differs from ours in that (1) the two embedding spaces in their model do not represent the same universe of objects; (2) the embeddings do not capture set relationships.

**Deep set representations**: Our paper is also about how to encode a set of objects with a neural network. One issue is how to ensure invariance to the order in which examples are presented. Vinyals et al. (2015) proposed an approach based on permutation-invariant content-based attention. For producing sets as outputs, Rezatofighi et al. (2017) proposed a probabilistic model, within a supervised learning paradigm where all classes are known at training time, that predicts both the cardinality and the particular elements of the set.

# 3   MODEL I: EMBEDDING $f$ AND COMPOSITION $g$

Given two examples $x_a$ and $x_b$ that are associated with singleton sets $\{s\}$ and $\{t\}$, respectively, the hope is that, for some third example $x_c$ that is associated with *both* classes (i.e., $\{s, t\}$), we have

$$g(f(x_a), f(x_b)) \approx f(x_c)$$

Moreover, we hope that $g$ can generalize to *any* number of classes within the set $\mathcal{S}$. For example, if example $x_d$ is associated with a singleton set $\{u\}$, then we hope

$$g(g(f(x_a), f(x_b)), f(x_d)) \approx f(x_e)$$

where $x_e$ is an example associated with $\{s, t, u\}$.

There are two challenging tasks that $f$ and $g$ must solve cooperatively: (1) $f$ has to learn to perceive multiple objects that appear simultaneously and may possibly interact with each other – all *without* knowing the rendering process $r$ of how examples are formed or how classes are combined. (2) $g$ has to define geometrical structure in the embedding space to support set union operations. One way to understand our computational problem is the following: If $f$ is invertible, then ideally we would want $g$ to compute $g(e_\mathcal{T}, e_\mathcal{U}) = f(r(c(f^{-1}(e_\mathcal{T})) \cup c(f^{-1}(e_\mathcal{U}))))$. In other words, one (though not necessarily the only) way that $g$ can perform well is to learn to perform the following actions (without knowing $r$ or $c$): (1) invert each of the two input embeddings; (2) classify the two corresponding label sets; (3) render an example with the union of the two inferred label sets; and (4) embed the result.

**One-shot learning**: Model I can be used for one-shot learning on a set of classes $\mathcal{S}$ not seen during training in the following way: We obtain $k$ labeled examples $x_1, \ldots, x_k$ from the user, where each $\{s_i\} = c(x_i)$ is the singleton set formed from the $i$th element of $\mathcal{S}$ and $|\mathcal{S}| = k$. We call these examples the *reference examples*. We then infer which set of classes is represented by a new example

$x'$ using the following procedure: (1) Compute the embedding of $x'$, i.e., $f(x')$. (2) Use $f$ to compute the embedding of each singleton example $x_i$, i.e., $e_{\{i\}} = f(x_i)$. (3) From $e_{\{1\}}, \ldots, e_{\{k\}}$, estimate the embedding of *every* subset $\mathcal{T} = \{s_1, \ldots, s_l\} \subseteq \mathcal{S}$ according to the recurrence relation:

$$e_{\{s_1, \ldots, s_l\}} = g(e_{\{s_1, \ldots, s_{l-1}\}}, e_{\{s_l\}}) \tag{1}$$

Finally, (4) estimate the label of $x'$ as

$$\underset{\mathcal{T} \subseteq \mathcal{S}}{\arg\min} |f(x') - e_{\mathcal{T}}|_2^2 \tag{2}$$

Although the number of possible subsets is exponential in $|\mathcal{S}|$, for speaker diarization the number of overlapping speakers is typically small, and thus the iteration is tractable.

## 3.1 TRAINING PROCEDURE

Functions $f$ and $g$ are trained jointly: For each example $x$ associated with classes $\mathcal{T}$, we compute $e_{\mathcal{T}}$ from the singleton prototypes according to Equation 1. (To decide the order in which we apply the recursion, we define an arbitrary ordering over the elements of $\mathcal{S}$ and iterate accordingly.) We then compute a hinge loss:

$$|f(x) - e_{\mathcal{T}}| \le |f(x) - e_{\mathcal{T}'}| - \epsilon$$

for every $\mathcal{T}' \ne \mathcal{T} \subseteq \mathcal{S}$, where $\epsilon$ is a small positive real number. In practice, for each example $x$, we randomly pick one element of $\mathcal{T}' \in 2^{\mathcal{S}}$ for comparison. See Appendix for a discussion of an alternative (but less effective) training procedure.

## 3.2 EXPERIMENT 1: SIMULATED 1-D AUDIO WAVEFORMS

To explore the viability of Model I, we first conducted a simulation using 1-D "audio" signals, in which each "speaker" $s_i \in \mathcal{S}$ is modeled with a prototype $p_i \in \mathbb{R}^n$ consisting of the superposition of some randomly chosen frequencies and phases. Then, the simultaneous sound from multiple (up to 3) speakers is given by a rendering function $r$ that computes a vector sum of slightly perturbed versions of the prototype vectors and then clips the result. (See Figure 2(a) for example waveforms, and the Appendix for more details.) We trained an embedding and composition functions $f$ and $g$, as described in Section 3.1, on a set of 250 speakers. We then tested these functions on test speakers (and examples) not seen during training after providing just a single "reference" example of each speaker for one-shot learning. The goal is to identify the exact set of speakers who contributed to the formation of each audio waveform.

**Architecture**: For function $f$ we used a simple convolutional neural network that produces unit-length vectors in $\mathbb{R}^{32}$:

$$\mathrm{BN} - \mathrm{Conv}(3, 32) - \mathrm{ReLU} - \mathrm{BN} - \mathrm{Conv}(3, 32) - \mathrm{ReLU} - \mathrm{MP}(2) - \mathrm{FC}(32) - \mathrm{ReLU} - \mathrm{FC}(32) - \mathrm{L2Norm}$$

where $\mathrm{Conv}(k, f)$ is a 1-D convolutional layer with $f$ filters of size $k$, stride 1, and zero-padding to preserve the spatial extent of the example; BN is batch normalization; $\mathrm{MP}(k)$ is a max-pooling layer with width and stride $k$; $\mathrm{FC}(n)$ is a fully-connected layer with $n$ neurons; and L2Norm is a $L_2$-normalization layer. We constructed the composition function ($g_{\mathrm{DNN}}$) to be symmetric:

$$\mathrm{Symm}(a, b; 32) - \mathrm{BN} - \mathrm{ReLU} - \mathrm{FC}(32) - \mathrm{BN} - \mathrm{ReLU} - \mathrm{FC}(32) - \mathrm{BN} - \mathrm{ReLU} - \mathrm{FC}(32) - \mathrm{L2Norm}$$

where $\mathrm{Symm}(a, b; k) = W_1 a + W_1 b + W_2(a \odot b)$ is a symmetric function (with parameter matrices $W_1, W_2$) of its two examples $a, b \in \mathbb{R}^n$ that produces a vector in $\mathbb{R}^k$.

**Assessment**: We evaluated accuracy (% correct) in multiple ways: (1) the accuracy, over all examples, of identifying the exact set of speakers; (2) the accuracy, separately for each set size (1,2,3), of identifying the exact set of speakers; and (3) the accuracy, over all examples, in distinguishing just the *number* of speakers in the set rather than their specific identities. As a baseline, we used an oracle that returned the Most Frequently (MF) occurring label set in the test set (or a random selection among multiple most frequent elements). **Results** are shown in Figure 2(b). Note that, for the rows that evaluate on $k$-sets, the embedding functions are still used to compare the test example to *every* possible subset (see Equation 2) – not just subsets of size $k$. In general, the compositional embedding method was far more accurate (48.1% for top-1 and 69.6% for top-3 on all subsets) compared to the baseline. The accuracy improvement was largest on singletons (1-sets), which made up 1/3 of the

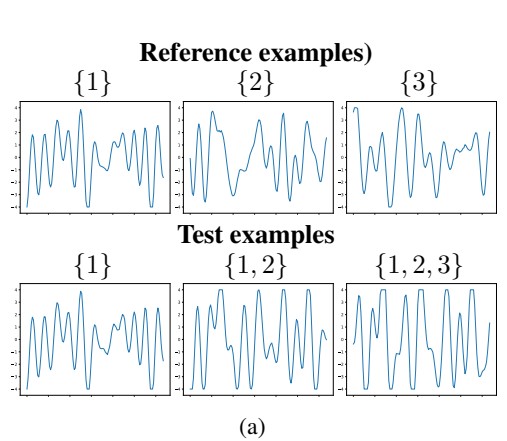

**Reference examples)**

{1}   {2}   {3}

**Test examples**

{1}   {1, 2}   {1, 2, 3}

(a)

| Experiment 1 (1-D Waveforms): Accuracy | | | |
|---|---|---|---|
| **Label Set Identification** | | | |
| | | $f$ & $g_{\text{DNN}}$ | MF |
| All | Exact | 48.1 (7.3) | 7.3 (0.2) |
| | Top-3 | 69.6 (8.6) | 21.7 (0.7) |
| 1-sets | Exact | 94.6 (5.9) | – |
| | Top-3 | 100.0 (0.0) | – |
| 2-sets | Exact | 33.7 (13.8) | – |
| | Top-3 | 88.5 (10.5) | – |
| 3-sets | Exact | 16.0 (9.2) | – |
| | Top-3 | 73.4 (13.5) | – |
| **Set Size Determination** | | | |
| All | | 61.9 (7.2) | 33.3 (0.0) |

(b)

Figure 2: **Experiment 1. (a)**: Simulated audio signals. Each class is generated by a prototype waveform. Each example is a noisy superposition of one or more prototypes. "Reference" examples are used for one-shot learning; "test" examples are used for evaluation. **(b)**: Mean (std.dev.) accuracy (% correct) in inferring the label sets exactly (top 1), within the top 3, and the size of each label set. Results are averaged over 50 test sets on classes not seen during training (one-shot learning).

test data, and for which any non-compositional embedding functions (i.e., $f$ trained without any $g$) would likely work well. Even on 2-sets and 3-sets, however, the compositional embedding approach was substantially more accurate than the baseline. Moreover, part of the task involves *inferring* the number of elements in the label set (set size determination); on this problem too the compositional embedding was nearly twice as accurate (61.9% vs. 33.3%) compared to the MF baseline.

**Geometry of the embedding space**: We visualized (using PCA) the embedding space for a set of 3 classes not used for training (Figure 1(b)). Each marker (the symbol/color combination distinguishes the different label sets) represents the embedding of a test example. The clusters of markers for each label set are clearly distinguished by the embedding $f$. From the examples from each label set, one is randomly chosen as the "reference" examples for 1-shot learning. Using composition function $g_{\text{DNN}}$, their embeddings are combined to estimate where, in the embedding space, an example for the union of their two label sets would lie; the estimated location is shown at the head of the arrow whose two tails come from the two reference examples. Although the estimated and actual clusters do not align exactly, there is substantial agreement. For instance, the examples for speakers $\{1, 2, 3\}$ (i.e., all 3 people talking simultaneously) are represented by the blue triangles pointing to the right, and the estimate, given by $f$ and $g_{\text{DNN}}$ from reference examples of $\{3\}$ and $\{1, 2\}$ is shown at the head of the arrow with label $\{1, 2, 3\}$ beside it. Not all compositions are accurate, e.g., the estimated location for label set $\{1, 3\}$ is somewhat below the actual cluster location.

### 3.3 EXPERIMENT 2: OMNIGLOT

We also evaluated our method on the OmniGlot dataset (Lake et al., 2015). OmniGlot contains handwritten characters from 50 different alphabets; in total it comprises 1623 symbols, each of which was drawn by 20 people and rendered as a 64x64 image. OmniGlot has been previously used in one-shot learning research (e.g., Rezende et al. (2016); Bertinetto et al. (2016)). In our experiment, the model is provided with one reference image for each singleton test class. Then, it uses $f$ and $g$ to select the subset of classes that most closely match the embedding of each test example (Equation 2). In this study we considered class label sets up to size 2 (i.e., singletons and 2-sets). The rendering function $r$ randomly picks one of the 20 exemplars from each class; it randomly shifts, scales, and rotates it; and then it adds it to an image with Gaussian noise (see Figure 3(a)). For images with multiple classes, the pixel-wise minimum across all classes is applied before adding the noise. (See Appendix for more details.) Similar to Experiment 1, the goal is to train $f$ and $g$ so that, on classes not seen during training, the exact set of classes contained in each test example can be

inferred. During both training and testing, each of the 15 class label sets (5 singletons and $\binom{5}{2} = 10$ 2-sets) occurred with equal frequency. All the embeddings were normalized to unit length.

**Architecture**: For $f$, we used ResNet-18 (He et al., 2016) that was modified to have 1 input channel and a 32-dimensional output. For $g$, we tested several architectures of increasing complexity: (a) **Bi-linear**: $g_{\text{Lin}} = \text{Symm}(a, b; 32) - \text{L2Norm}$. (b) **Bi-linear + FC**: $g_{\text{Lin+FC}} = \text{Symm}(a, b; 32) - \text{BN} - \text{ReLU} - \text{FC}(32) - \text{L2Norm}$. (c) **DNN**: $g_{\text{DNN}}$ is defined as in Section 3.2.

**Training**: The process is similar to Experiment 1. See Appendix for details. **Testing**: Similar to training, the testing data are grouped into 5 randomly selected classes (not seen during training), and images from these classes are rendered using function $r$ from either singleton or 2-set class label sets. We optimize Equation 2 to estimate the label set for each test example.

**Baselines**: We compared to several baselines:

- **Most frequent (MF)**: Always guess the most frequent element in the test set. Since all classes occurred equally frequently, this was equivalent to random guessing.

- **Traditional $f$ with simulated composite reference examples (SimRef)**: Given the reference examples of the singleton label sets (e.g., $\{1\}, \{3\}$), we can simulate reference examples for each composite label set (e.g., $\{1, 3\}$) and then use a traditional embedding approach to classify the test examples. We simulated imperfect knowledge of the rendering function $r$ by performing pixel-wise minimum, but *without* shifting/scaling/rotation. In other words, this approach simply superimposes pairs of reference images on top of each other to simulate composite examples. To estimate the label set of each test example, we select the reference example (including the simulated composite references) with minimal $L_2$ distance.

- **Traditional $f$ and average (Mean)**: $f$ is trained as a traditional embedding function (only on singletons) using one-shot learning (Koch et al., 2015). Then the embedding of a composite image is computed using the mean of the embeddings of each class in its label set.

- **Traditional $f$ trained jointly with $g_{\text{mean}}$ ($g_{\text{mean}}$)**: $f$ is trained on only singleton embeddings but is optimized jointly with composition $g_{\text{mean}}$ that computes the mean of its inputs. The difference with the previous baseline is that $f$ can benefit from knowing how its embeddings are combined.

**Results**: As shown in Figure 3(b), the proposed $f$ & $g$ method outperforms the Mean and $g_{\text{mean}}$ baselines. Upon closer investigation, we discovered that, while Mean can distinguish each singleton from the other four singletons with high accuracy (0.979, not shown in table), it struggles when it must decide among all 15 possible label sets. The slightly more powerful $f$ & $g_{\text{Mean}}$ method (i.e., the composition function has no trainable parameters) can achieve better accuracy on 2-set images, but the accuracy on 1-set images was much worse than other methods. We argue that $f$ and $g$ correspond to two different tasks, and the model is not able to do a good job on both tasks at the same time without optimizing $g$. If $g$ is a learnable function, then a simple symmetric linear function ($g_{\text{Lin}}$) achieves the best performance. After we stack more FC layers, the performance gets worse. The reason could be overfitting, and it may be possible to achieve better performance with regularization or more training data.

## 4  MODEL II: EMBEDDING $f$ AND QUERY FUNCTION $h$

In real life, it is very difficult to create a dataset that all singletons in a compositional set are labeled. Thus, we want to extend our compositional embedding from "composition" to "containing". Here we consider a second type of compositional embedding mechanism that tests whether the set of classes associated with one example subsumes the set of classes associated with another example. We implement this using a "query" function $h$ that takes two embedded examples as inputs:

$$h(f(x_a), f(x_b)) = \text{True} \iff c(x_b) \subseteq c(x_a) \tag{3}$$

In contrast to $g$, function $h$ is not symmetric: its first and second arguments are the putative superset and subset, respectively. Also, $h$ can be trained in an *unsupervised* manner w.r.t. the individual examples: it never needs to know which particular label set is associated with an example, but rather only *pairwise* information about which examples "subsume" other examples. For $h$, we tried several functions ($h_{\text{DNN}}, h_{\text{Lin+FC}}, h_{\text{Lin}}$), analogous to the different implementations of $g$ from Section 3.3. The final layers of all models have a 1-dimensional output.

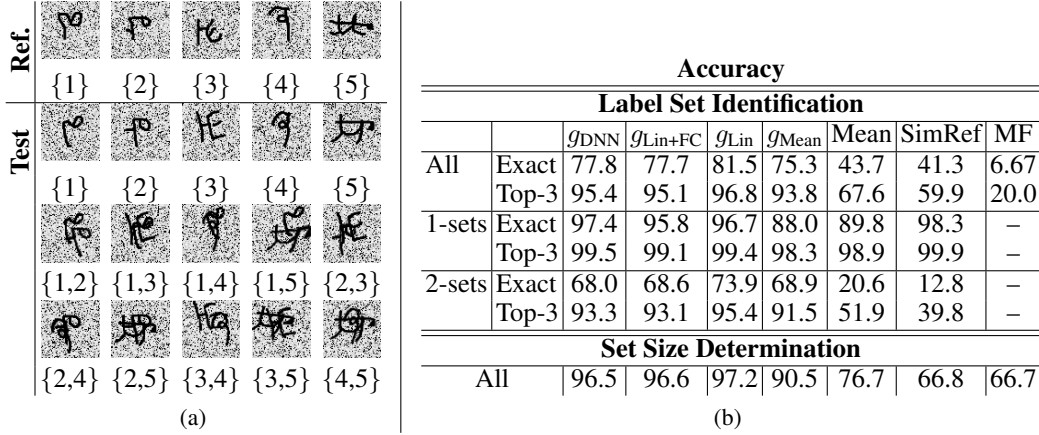

| | | | Accuracy | | | | | | |
|---|---|---|---|---|---|---|---|---|---|
| | | **Label Set Identification** | | | | | | | |
| | | $g_{DNN}$ | $g_{Lin+FC}$ | $g_{Lin}$ | $g_{Mean}$ | Mean | SimRef | MF |
| All | Exact | 77.8 | 77.7 | 81.5 | 75.3 | 43.7 | 41.3 | 6.67 |
| | Top-3 | 95.4 | 95.1 | 96.8 | 93.8 | 67.6 | 59.9 | 20.0 |
| 1-sets | Exact | 97.4 | 95.8 | 96.7 | 88.0 | 89.8 | 98.3 | – |
| | Top-3 | 99.5 | 99.1 | 99.4 | 98.3 | 98.9 | 99.9 | – |
| 2-sets | Exact | 68.0 | 68.6 | 73.9 | 68.9 | 20.6 | 12.8 | – |
| | Top-3 | 93.3 | 93.1 | 95.4 | 91.5 | 51.9 | 39.8 | – |
| | | **Set Size Determination** | | | | | | | |
| All | | 96.5 | 96.6 | 97.2 | 90.5 | 76.7 | 66.8 | 66.7 |

(a)      (b)

Figure 3: **Experiment 2. (a)**: Reference and test examples of OmniGlot images. **(b)**: Mean accuracy (% correct) in inferring the label set of each example exactly (top 1), within the top 3, and the size of each label set. Results are computed on classes not seen during training (one-shot learning).

| **Experiment 3 (OmniGlot)** | | | | |
|---|---|---|---|---|
| | $h_{DNN}$ | $h_{Lin+FC}$ | $h_{Lin}$ | TradEmb |
| % Correct | 77.9 | 77.3 | 73.8 | 75.3 |
| AUC | 86.8 | 85.5 | 81.7 | 85.6 |

(a)

| **Experiment 4 (COCO)** | | | | | |
|---|---|---|---|---|---|
| | $h_{DNN}$ | $h_{Lin+FC}$ | $h_{Lin}$ | TradEmb | ResNet |
| % Correct | 64.8 | 62.5 | 62.1 | 54.8 | 58.4 |
| AUC | 71.2 | 67.7 | 67.3 | 57.2 | 58.7 |

(b)

Table 1: Accuracy to query whether the classes in one example subsume the classes in another example, for the proposed $f$ & $h$ method versus a "traditional" embedding function $f$ as a baseline. Method $f$ & $h$ is the proposed model; TradEmb is a baseline. **(a)**: Experiment 3. **(b)**: Experiment 4.

## 4.1 TRAINING PROCEDURE

Functions $f$ and $h$ are trained jointly. Since $h$ is not symmetric, its first layer is replaced with two weight matrices for the different input embeddings (see Appendix). In contrast to Model I, "reference" examples are not needed; only the subset relationships between label sets of pairs of examples are required. Model II can be trained on one set of classes and applied to a different set of classes (see Experiment 3), akin to zero-shot learning. It is also useful when the training and testing classes are the same because, in contrast to traditional supervised training of a detector for each possible class, no labels for individual examples are needed. To train $f$ and $h$, we backpropagate a binary cross-entropy loss, based on correctly answering the query in Eq. 3, through both $f$ and $h$.

## 4.2 EXPERIMENT 3: OMNIGLOT

Here we assess Model II in a one-shot learning setting (OmniGlot), i.e., training and testing classes are different. As in Experiment 2, we consider class label sets of size up to 2, and we use the same rendering function $r$. Let $f(x_a)$ and $f(x_b)$ be the second arguments to $h$. For $x_a$, we always choose examples associated with two classes, i.e., $c(x_a) = \{s_1, s_2\}$. For $x_b$, half of the positive examples (i.e., such that $h(f(x_a), f(x_b)) = $ True) contain either $s_1$ or $s_2$, and half contain both classes. For the negative examples ($h(f(x_a), f(x_b)) = $ False), $x_b$ is associated with some other singleton class $s_3 \notin \{s_1, s_2\}$. See Appendix for more details. **Baseline**: We compared our proposed method with a traditional (non-compositional) embedding method (**TradEmb**) that is trained to separate examples according to their association with just a *single* class. In particular, for each composite example $x_a$ (i.e., $|c(x_a)| = 2$), we picked one of the two classes arbitrarily (according to some fixed ordering on the elements of $\mathcal{S}$); call this class $s_1$. Then, we chose both a positive example $x_b$ (such that $c(x_b) = \{s_1\}$) and a negative example $x_c$ (such that $c(x_c) = \{s_3\} \not\subset c(x_a)$). We then compute a hinge loss so that the distance between $f(x_a)$ and $f(x_b)$ is smaller than the distance

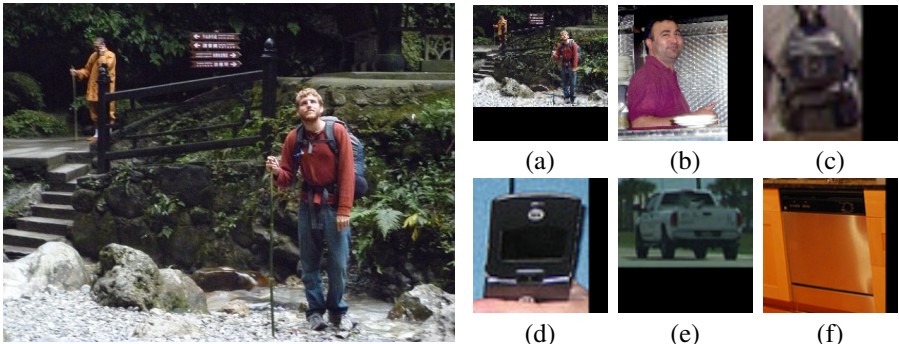

Figure 4: **Experiment 4 (COCO)**: An example image (left) of two people, one talking on a cell phone, and the other wearing a backpack. The image is padded to form a square and then downscaled (a). The composite embedding ($f$ & $h$) is computed and then queried about the presence of the object in images (b), (c), (d), (e), and (f) containing *person*, *cell phone*, *backpack*, *truck*, and *oven*, respectively. The query function $h$, when given the embeddings of image (a) and another image, should return True for (b),(c),(d) and False for (e),(f).

between $f(x_a)$ and $f(x_c)$, and backpropagate the loss through $f$. During testing, we use $f$ to answer a query – does $c(x_a)$ contain $c(x_b)$? – by thresholding the distance between $f(x_a)$ and $f(x_b)$ (threshold of 0.5). **Results** are shown in Table 1(a). Compositional embeddings, as implemented with a combination of $f$ trained jointly with either $h_{\text{DNN}}$, $h_{\text{Lin+FC}}$, or $h_{\text{Lin}}$, slightly outperformed the TradEmb baseline, in terms of both % correct accuracy and AUC. Moreover, there was an advantage of deeper architectures for $h$.

## 4.3 EXPERIMENT 4: COCO

We trained and evaluated Model II on COCO (Lin et al., 2014). This is a highly challenging problem: in the example in Figure 4, $f$ has to encode a backpack, a cell phone, and a person; then, given completely different images of these classes (and others), $h$ has to decide which objects were present in the original image. Here, training and testing classes are the same (but testing examples are not used for training). In COCO, each image may contain objects from multiple classes, and each object has a bounding box. We used the bounding boxes to crop singleton objects from images. For training, we used the same strategy as Experiment 3, except that in positive queries (i.e., pairs of images $x_a, x_b$ such that $h(f(x_a), f(x_b)) = \text{True}$), image $x_b$ was always associated with just a singleton label set. During testing, half of $x_b$ are contained in $x_a$ and half are not (see Appendix). **Baseline**: We compared to the TradEmb method: to decide the singleton label set for each image during training, we pick the largest object. Since in this problem the training and testing classes are the same, we also implemented another baseline (see Appendix) consisting of a ResNet classifier with multiple independent binary outputs (one for each class). We can then answer queries about the label sets of two images using the labels of the different classes estimated by the ResNet for each image. **Results** are shown in Table 1(b). The proposed methods ($f$ & one of the $h$ functions) easily outperformed TradEmb. They also outperformed the ResNet baseline. One possible reason for the latter baseline's poor performance was that the data was highly imbalanced (since most images contain just a few images). As in Experiment 3, deeper architectures for $h$ performed better.

## 5 CONCLUSIONS

We proposed a new kind of embedding mechanism whereby the *set* of objects contained in the input data (e.g., image, video, audio) must be both *perceived* and then mapped into a space such that the *set relationships* – union (Model I) and subset (Model II) – between multiple embedded vectors can be inferred. Importantly, the ground-truth rendering process for how examples are rendered from their component classes is not known and must implicitly be learned. In our experiments, conducted on simulated data, OmniGlot, and COCO, the accuracy was far from perfect but outperformed several baselines, including one based on a traditional embedding approach. The results provide a proof-of-

concept of how an embedding function $f$, trained jointly with either the composition function $g$ or the query function $h$, could be effectively optimized. One possible direction for further research to increase accuracy is to take better advantage of the statistical structure of class co-occurrence in a specific application domain (e.g., which objects tend to co-occur in the same image).

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

## A    ALTERNATIVE TRAINING PROCEDURE

We also tried another method of training $f$ and $g$ with the explicit goal of encouraging $g$ to map $e_{\mathcal{T}}$ and $e_{\mathcal{U}}$ to be close to $e_{\mathcal{T} \cup \mathcal{U}}$. This can be done by training $f$ and $g$ alternately, or by training them jointly in the same backpropagation. However, this approach yielded very poor results. A possible explanation is that $g$ could fulfill its goal by mapping all vectors to the same location (e.g., $\mathbf{0}$). Hence, a trade-off arises between $g$'s goal and $f$'s goal (separating examples with distinct label sets).

## B    EXISTENCE OF $f$ & $g$: ALGEBRAIC CONSTRUCTION

Since it was initially not clear to us whether functions $f$ and $g$ even exist with the desired properties that could plausibly be implemented as a neural network, we first sought to design them analytically. Composition function $g$ needs to map two vectors on a unit sphere back onto the same sphere, so that the *location* on the sphere uniquely identifies a set of classes. Here, we assume the set of classes is the same for training and testing. Our construction is inspired by basic group theory, in particular how each element of a permutation group can be represented as a rotation of Euclidean space and performed by multiplication with a permutation matrix.

Let $|\mathcal{S}| = k$. We model $g$ after the permutation group of degree $m = 2k$ consisting of all pairwise exchanges of coordinates $2i$ and $2i - 1$ ($i = 1, \ldots, k$), where the group action is composition. Since each such permutation is independent of all the others, the group is commutative and contains contains $2^k$ elements – just as we desire so that the range of $g$ contains $2^{|\mathcal{S}|}$ elements. Each member of the permutation group can be associated with a unique permutation matrix, where multiplication among these matrices is also commutative. Define a vector $e_{\emptyset} \in \mathbb{R}^{2k}$ whose length is 1 and whose components are all distinct. The first condition is standard when creating an embedding, and the second is important so that the permutations applied to the vector can be deduced unambiguously. We associate vector $e_{\emptyset}$ with the empty set $\emptyset$. Next, to each singleton $\{s_i\}$, where $s_i \in \mathcal{S}$, we associate the $m$-dimensional permutation matrix $P_{\{s_i\}}$ that swaps axes $2i$ and $2i - 1$.

We define $f$ as a neural network: In its first layers, all the classes associated with example $x$ are detected independently using $k$ binary classifiers. Next, for all the detected classes, their associated permutation matrices $P$ are multiplied together, which yields another commutative permutation matrix – the resultant matrix identifies all the classes present in the example. Finally, the result is multiplied by vector $e_{\emptyset}$. This produces a vector with unit length in the embedding space $\mathbb{R}^m$. If the first few layers comprise $k$ binary classifiers $d_i(z) \in \{0, 1\}$ ($i = 1, \ldots, m$), then the whole $f$ network computes:

$$f(x) = \left( \prod_{i : s_i \in \mathcal{S}} \left( P_{\{s_i\}} \right)^{d_i(z)} \right) e_{\emptyset}$$

We then construct $g$ so that, given two embedded vectors $e_{\mathcal{T}}, e_{\mathcal{U}}$, it computes the product of their associated permutation matrices and multiplies the result by $e_{\emptyset}$:

$$g(e_{\mathcal{T}}, e_{\mathcal{U}}) = g(P_{\mathcal{T}} e_{\emptyset}, P_{\mathcal{U}} e_{\emptyset}) \doteq P_{\mathcal{T}} P_{\mathcal{U}} e_{\emptyset} = P_{\mathcal{T} \cup \mathcal{U}} e_{\emptyset}$$

This construction of $f$ and $g$ enables perfect inference of the union of classes associated with any examples, provided the binary classifiers are perfect. This gave us hope that compositional set embeddings could succeed.

## C    DETAILS OF EXPERIMENT 1

Each speaker was represented by a "prototype" waveform comprising the sum of 8 sinusoids (range $[-1, +1]$) with randomly chosen frequencies and phases. To construct each training and testing example, the prototypes of all the speakers contained in the particular example are added together and then clipped at $[-4, +4]$. To simulate noise, we slightly perturbed the frequencies and phases of each prototype by random noise drawn uniformly from $[-0.02, +0.02]$. We also perturbed the superposition by a small amount (from $[-0.02, +0.02]$) prior to clipping. The training dataset contained a total of 250 simulated speakers. Each minibatch comprised 3072 examples from 5 unique speakers that were combined in randomly selected subsets $\mathcal{T}$: 1024 of these contained individual speakers (singletons), 1024 contained combinations of 2 speakers, and 1024 contained combinations of 3 speakers. Functions $f$ and $g$ were trained jointly as described in Section 3.1. Training was conducted for 4000 epochs with a learning rate of 0.002 and the Adam algorithm. We set the hinge parameter $\epsilon = 0.1$. At test time, one of the singleton examples was randomly chosen as the "reference" example for one-shot learning. Similar to training, each test set consists of 3072 examples comprising randomly chosen 1-sets, 2-sets and 3-sets of speakers; see Figure 2(a) for example waveforms.

## D    DETAILS OF EXPERIMENT 2

To generate an example image from a class prototype, we applied random affine transformations consisting of shift up to $20\%$, scaling up to $10\%$, and rotation up to $10°$. Gaussian noise was added with mean 0.9 and variance 0.1. We set the hinge parameter $\epsilon = 0.1$.

Training is performed on 964 OmniGlot classes; from each of these, the 20 images are augmented using affine transformations (described above) to yield 50 images. Each mini-batch comprises 10000 examples from 5 classes, with two images randomly selected per class: one as a reference example, and one as a training example. The validation set consists of 20 OmniGlot classes not used for training. Training is performed using Adam (lr $= .0003$) to maximize the validation accuracy, up to a maximum of 100 epochs. Testing accuracy is computed over the 659 OmniGlot classes not used for training or validation.

## E    DETAILS OF EXPERIMENT 3

The dataset used in this experiment is the same as experiment 2 including the distribution of training set, validation set and test set.

We also used the same ResNet-18 as for Experiment 2. Function $h$ was constructed the same as $g$ except that (1) the first layer was relaxed to be asymmetric: $W_1 e_{\mathcal{T}} + W_2 e_{\mathcal{U}}$, and the output layer is logistic sigmoid. For training and testing, the proportion of positive and negative queries was fixed to be equal. During training, each mini-batch contains 128 query pairs and each query pair is composed by a positive query and a negative query. We use binary cross entropy loss and Adam optimizer (lr $= .0003$) to optimize the models. During evaluation, 0.5 is used for thresholding.

## F    DETAILS OF EXPERIMENT 4

COCO's training set is used for training, and COCO's labels of subclasses are used as training labels. A small part of images (100) are taken as validation set.

For ResNet baseline, We modified ResNet's last's dimension to 80 and applied logistic sigmoid to all of them. Thus, the output layer can be used as 80 classifiers and each one corresponds to one possible subclass.

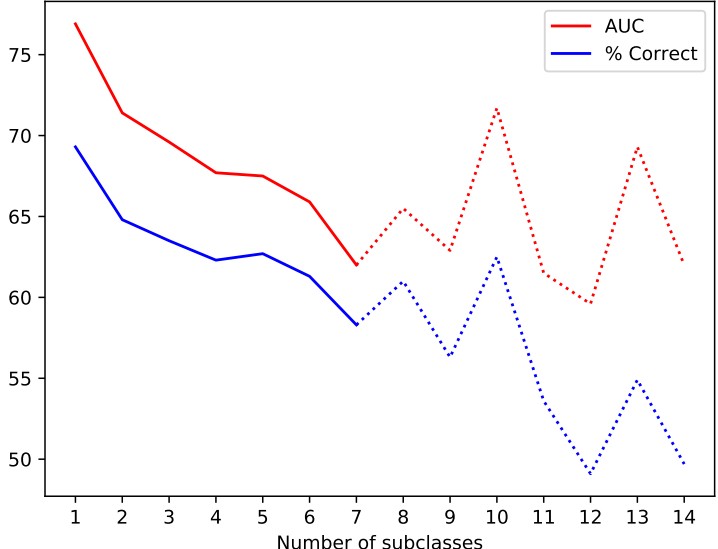

Figure 5: $h_{\textbf{DNN}}$**'s results according to the number of subclasses contained in images**: Results of images contain more than 7 labeled objects are showed by dotted lines because they have fewer than 100 samples in test set.

During training, the loss is composed by binary cross entropy loss from all 80 binary classifiers. For every subclass in all 80 possibilities, if it is in the set of subclasses of one image, then the label is 1, otherwise 0. Adam optimizer $((lr) = .0003)$ is used for optimization. During evaluation, for each query we assume the class labels of subclass images are already known.

ResNet is not optimized to answer queries about image pairs. Instead, it tries to encode each image into an n-bit string (for n classes). While this representation can account for all $2^n$ possible label sets, it may not be the most effective or efficient representation for the task, especially since some objects are very unlikely to co-occur with others. The proposed $f\&h$ embedding method can harness the co-occurrence structure to answer queries more accurately, whereas a ResNet trained to recognize all the individual classes does not harness it.

For other models singleton images are also required, which are cropped according to COCO's labels of bounding boxes. All images were padded with zeros to be square and then downscaled to 128x128.

We used the same architecture as for Experiment 3, except that the input layer was 3 channels instead of 1. Testing was the same as Experiment 3 (likewise with a distance threshold of 0.5). In both training and testing, the number of positive queries and negative queries are balanced.

In order to explore how the number of subclasses affects $h()$ function, we also plot the accuracy and AUC with different number of subclasses in Figure 5. We can see a decreasing trend of performance when the images contain more kinds of labeled objects.

