# OpenReview forum: "Compositional Embeddings: Joint Perception and Comparison of Class Label Sets"
_ICLR.cc/2020/Conference — Reject_

### Official Review · AnonReviewer1 · 2019-10-22
**Official Blind Review #1**

**Rating:** 6

**Review:**

Summary:
=======
This paper proposes compositional embeddings i.e. embeddings that can be used to infer multiple classes from the data. In particular, the paper deals with two types of composite functions for embeddings, one that computes union of the different classes represented by each embedding vector, and the other where the class of one of the embeddings is subsumed by the class of the other embedding. The actual composition functions are parameterized by neural networks whose parameters are learned from data. Results on synthetic as well as several real-world datasets highlight the superiority of the learned composite embeddings.



Comments:
==========
1) This paper presents a welcome contribution to the saturated literature on embeddings. The whole idea of compositionally and its application to speaker diarization and multi-object detection is novel.

2) The execution of the idea is also excellent and thorough. Further, the paper is very well written and puts itself nicely in context of previous work. I think this should inspire future work on other kinds of composite functions other than the two considered here.

3) The results on both the synthetic and real-world omniglot and COCO datasets are impressive and mostly well executed and show significant improvement over the "most frequent" baseline.


4) My only concern regarding the paper is w.r.t some arbitrary decisions made in the experiments e.g. how was the exact neural architecture for f in section 3.2 chosen? It seems contrived. Is it possible to do some ablation studies? Also, I think it will be nice to provide some more details regarding the neural network training in Section 3.1.


**Experience Assessment:**

I have read many papers in this area.

**Review Assessment: Checking Correctness Of Derivations And Theory:**

N/A

**Review Assessment: Checking Correctness Of Experiments:**

I carefully checked the experiments.

**Review Assessment: Thoroughness In Paper Reading:**

I read the paper at least twice and used my best judgement in assessing the paper.

---

> ### Author Response · Authors · 2019-11-15
> **More experiments have been conducted to answer the reviewer's concern**
>
> “how was the exact neural architecture for f in section 3.2 chosen? It seems contrived. Is it possible to do some ablation studies?” -- In our updated paper we compare models with different numbers of layers (g_Lin, g_Lin+FC, g_DNN). We also add some more details about training in the appendix.

---

### Official Review · AnonReviewer3 · 2019-10-23
**Official Blind Review #3**

**Rating:** 3

**Review:**

The authors propose a joint/compositional embedding procedure where a single instance can be mapped/embedded to multiple classes while preserving the class-specific information in the embedded representations. The authors look at class union and class query criteria for the composite embeddings. The proposed approach is evaluated appropriately. There are several issues with the work.

Does the proposal mean each embedding eventually corresponds to multiple classes/subclasses ie., one can learn something on-trivial about each class from these embeddings that is different from class-specific embedding? How do you avoid the trivial solution problem here i.e., the embeddings are going to be average of the class-specific embeddings --- as we see in the evaluations this is in fact happening (figure 1b)? Also, is this behaviour desired i.e., tending towards mean?

And continuing along these lines, a clear choice of baseline for the proposal is to choose mean embeddings i.e., men of independent embeddings? Or is this not appropriate? Why is ML the best baseline? We can use the probability map (the input to final softmax) instead as the embedding as well correct?

"... x_a containing objects in another image " -- this statement is not making sense, is it objects in x_a also present in another image x_b?

It is rather difficult to interpret the usefulness of g(.) when it is a nonlinear model like neural network. Simpler models (like Symm(a,b,.) i.e., just the first layer of what is being used now) should be evaluated instead to get better understanding of what is going on!


**Experience Assessment:**

I have published one or two papers in this area.

**Review Assessment: Checking Correctness Of Derivations And Theory:**

I assessed the sensibility of the derivations and theory.

**Review Assessment: Checking Correctness Of Experiments:**

I carefully checked the experiments.

**Review Assessment: Thoroughness In Paper Reading:**

I read the paper thoroughly.

---

> ### Author Response · Authors · 2019-11-15
> **Paper updated with new experiments and clarification**
>
> “Does the proposal mean each embedding eventually corresponds to multiple classes/subclasses ie., one can learn something on-trivial about each class from these embeddings that is different from class-specific embedding?” — Yes, that is the goal. The embedding computed by f can encode an entire *set* of classes, not just 1 class (as with traditional embeddings).
>
> “How do you avoid the trivial solution problem here i.e., the embeddings are going to be average of the class-specific embeddings” — Based on the reviewer’s suggestion, we added several more comparisons in Experiments 2, 3, and 4. In particular, we compared our proposed method to (1) “Mean”: Simply computing the mean of multiple embeddings from an embedding function f trained just on singletons. (2) “f & g_mean”: Computing the mean of multiple embeddings when the embedding f was trained *with the knowledge* that its outputs would be averaged together. In summary: we found evidence that the proposed method, based on f and a non-linear g, can deliver better performance than either of the two “mean” baselines.
>
> “‘... x_a containing objects in another image ‘ -- this statement is not making sense, is it objects in x_a also present in another image x_b?” -- Yes, that is correct. Objects in x_a are presented in x_b.
>
> “Simpler models (like Symm(a,b,.) i.e., just the first layer of what is being used now) should be evaluated instead to get better understanding of what is going on.” -- Thanks for the suggestion. We have implemented several new variants of g (and of h) in our updated paper. In some cases, a simple g consisting of a single linear layer works best, whereas in other cases a deeper g works better. Please note that we also fixed a bug in the implementation of the bi-linear baseline from our original submission. The result (which is now called the g_Lin method) has been updated in the paper. In experiment 3, we also used a different random seed and the new results are slightly different from the previous version.

---

### Official Review · AnonReviewer2 · 2019-10-24
**Official Blind Review #2**

**Rating:** 6

**Review:**

This paper describes a way to train functions that are able to represent the union of classes as well as to query if the classes in an image subsume the classes in another image. This is done throughly jointly training embedding functions, a set union function and a query function. The paper reads well.

While the approach is reasonable, the experiments seem to be quite incomplete and no explanation is given why a trivial solution cannot be used instead of the learnt functions.

The paper argues for learning a set union function however much of the evaluation focuses on quite small sets of 2 or 3 items. On the evaluation that utilises larger sets, e.g. COCO, there isn't any analysis of how performance of the technique scales with the size of the set since that would be one of the defining characteristics of a set union function. The COCO experiment is also lacking in detail, for example, how many items are there in the positive and negative sets and how the test set is balanced. Finally, it seems that f, g and h could be trivial non-learnt functions. For example, f could be a function that maps an image to a binary representation of its classes (this could be a typical ResNet image classifier), g could be a function that does a binary OR of its two arguments and h could be a function that uses a binary AND and equality test on its two arguments. In this case, g and h don't need to be learnt at all. This may not be possible in the COCO experiment where the individual labels are not known but it seems quite unrealistic to have a dataset where only pairwise subset relationships are known.

It also seems that the f is always different between that used with g and that used with h, is this the case? SimRef also doesn't do data augmentation but there's no explanation why it is done for the proposed method and not for this baseline. The MF baseline in experiment 1 seems to be a straw man especially since the baselines in experiment 2 are much stronger.

================================================================================
Update after rebuttal:

Thanks for answering my questions and performing the additional experiments with a ResNet baseline and performing an additional analysis based on the number of subclasses in figure 5. I think these provide a substantially better analysis of the algorithm so I've increased my score correspondingly. For the final paper, I think it would be good to add TradEmb/ResNet to figure 5 as well to understand how those methods scale worse/better with the number of subclasses.

**Experience Assessment:**

I have read many papers in this area.

**Review Assessment: Checking Correctness Of Derivations And Theory:**

I assessed the sensibility of the derivations and theory.

**Review Assessment: Checking Correctness Of Experiments:**

I assessed the sensibility of the experiments.

**Review Assessment: Thoroughness In Paper Reading:**

I read the paper at least twice and used my best judgement in assessing the paper.

---

> ### Author Response · Authors · 2019-11-15
> **We would like to thank the reviewer for noting some missing points in our experiments. We updated the paper with some new experiments according to the suggestions and made some clarification.**
>
> “No explanation is given why a trivial solution cannot be used instead of the learnt functions.” — First, we want to point out that training a classifier (e.g., ResNet) using standard supervised learning is only possible if the training and testing classes are the same. For our Omnigot and simulation studies (Experiments 2 and 1, respectively), they were different (one-shot learning). This is an important case, e.g., for speaker diarization. Second, based on the reviewer’s suggestion, we did conduct a follow-up analysis on COCO (Experiment 4, in which training and testing classes are indeed the same) -- please see the updated paper. Interestingly, the trained ResNet classifier (followed by a threshold of 0.5 and then a bit-comparison to answer label queries) did not perform very well compared to the proposed f & h method -- see Table 1(b). One possible reason is that ResNet is not optimized to answer queries about image pairs. Instead, it tries to encode each image into an n-bit string (for n classes). While this representation can account for all 2^n possible label sets, it may not be the most effective or efficient representation for the task, especially since some objects are very unlikely to co-occur with others. The proposed f & h embedding method can harness the co-occurrence structure to answer queries more accurately, whereas a ResNet trained to recognize all the individual classes does not harness it. Another reason may be that such a classifier trained on COCO has to overcome strong class imbalance (which is not trivial to fix on COCO), which the compositional embeddings do not (since they were trained inherently with 50%/50% balance).
>
> “Analysis of how performance of the technique scales with the size of the set” — We added a study to the appendix on the accuracy of f & h as a function of the label set size.
>
> “f is always different between that used with g and that used with h, is this the case?” — f is the same architecture but has different parameters in g than h.
>
> “SimRef also doesn't do data augmentation but there's no explanation why…” — Actually, SimRef uses the same augmentation as the proposed f & g method. Recall that all the methods receive reference examples of the *singleton* classes, which are created using random affine transformations of the original OminGlot data. The reviewer may be referring to the statement, “without shifting/scaling/rotation” in our paper. Please note that these transformations were part of the *rendering* function r. Since r is assumed to be hidden (from all the methods), we did not give oracle access of how r works to the SimRef method.

---

### Comment · Area_Chair1 · 2019-11-15
**Reviewers, any comments on the author response?**

Dear Reviewers, thanks for your thoughtful input on this submission!  The authors have now responded to your comments.  Please be sure to go through their replies and revisions.  If you have additional feedback or questions, it would be great to know.  The authors still have one more day to respond/revise further.  Thanks!

---

### Decision · Program_Chairs · 2019-12-19

**Decision:**

Reject

**Comment:**

The authors propose a new type of compositional embedding (with two proposed variants) for performing tasks that involve set relationships between examples (say, images) containing sets of classes (say, objects).  The setting is new and the reviewers are mostly in agreement (after discussion and revision) that the approach is interesting and the results encouraging.  There is some concern, however, that the task setup may be too contrived, and that in any real task there could be a more obvious baseline that would do better.  For example, one task setup requires that examples be represented via embeddings, and no reference can be made to the original inputs; this is justified in a setting where space is a constraint, but the combination of this setting with the specific set query tasks considered seems quite rare.  The paper may be an example of a hammer in search of a nail.  The ideas are interesting and the paper is written well, and so the authors can hopefully refine the proposed class of problems toward more practical settings.